# Mechanical Transmission of Lumpy Skin Disease Virus by *Stomoxys* spp. (*Stomoxys calsitrans*, *Stomoxys sitiens*, *Stomoxys indica*), Diptera: Muscidae

**DOI:** 10.3390/ani10030477

**Published:** 2020-03-12

**Authors:** Arman Issimov, Lespek Kutumbetov, Mukhit B. Orynbayev, Berik Khairullin, Balzhan Myrzakhmetova, Kulyaisan Sultankulova, Peter J. White

**Affiliations:** 1Sydney School of Veterinary Science, Faculty of Science, University of Sydney, Sydney 2006, NSW, Australia; p.white@sydney.edu.au; 2RGE “Research Institute for Biological Safety Problems” Committee of Science, The Ministry of Education and Science of the Republic of Kazakhstan, Gvardeiskiy 080409, Kordaiskiy rayon, Zhambylskaya oblast, Republic of Kazakhstan; lespek.k@gmail.com (L.K.); omb65@mail.ru (M.B.O.); khirullin@mail.ru (B.K.); balzhan.msh@mail.ru (B.M.); sultankul70@mail.ru (K.S.)

**Keywords:** lumpy skin disease, stable flies, vector born disease, mechanical transmission

## Abstract

**Summary:**

Lumpy skin disease (LSD) is an emerging disease in Kazakhstan, and currently the means of transmission is uncertain. In the current study, mechanical transmission of lumpy skin disease virus (LSDV) by *Stomoxys* species from infected to naive animals was demonstrated under laboratory conditions. Flies partially fed on LSDV-infected cattle were placed onto recipient animals within a 1 h time period to complete their feeding process. In addition to this, virus was isolated from all three *Stomoxys* species immediately and 6 h post feeding on LSDV infected animal, while virus DNA was detectable up to 48 h post-feeding by PCR.

**Abstract:**

Samples collected for PCR from recipient animals tested positive in 5 out of 6 cases, while the virus was isolated from 4 of 6 animals. The clinical signs exhibited by recipient animals were mostly moderate in nature with only one severe case. To our knowledge, this is the first time that transmission of LSDV by three *Stomoxys* species has been demonstrated, and their role as mechanical vectors of LSDV is indicated.

## 1. Introduction

Lumpy skin disease virus (LSDV) belongs to the *Capripoxvirus* genus of the Poxviridae family and is a highly contagious infectious disease of cattle, mainly characterised by multiple skin lesions, fever, enlargement of superficial lymph nodes, profuse salivation, lacrimation and nasal discharge as well as oedema and swelling of the joints [1]. The disease was recorded in Zambia for the first time in 1929. Subsequently, LSDV has become endemic throughout the African continent and Near East and continues to spread north posing a threat to Europe and Central Asian regions. Recently, an LSD outbreak has been documented in Greece [2] and, within a month, the disease was clinically confirmed and reported in Azerbaijan, the Russian Federation and the Republic of Kazakhstan [3,4,5].

The World Organization for Animal Health (OIE) classified LSD as a notifiable disease due to the fact of its significant economic impact [6]. In addition, it has a detrimental effect on the development of animal production. Economic damage results from a sharp decline in milk yield, milk quality and hide damage, body weight reduction, abortion, infertility and in some cases death of the animal [7]. Morbidity rates may vary significantly during LSD outbreaks and reach up to 100%, whereas the mortality rate is usually low (less than 5%), reaching 20% on some occasions [6,8,9].

It is thought that a variety of blood-feeding insects may play a significant role in LSDV transmission by acting as mechanical vectors. This assumption is based on the seasonality of outbreaks of LSD, occurring during hot and wet summer seasons [10,11,12]. Studies on using the basic reproduction numbers to evaluate the risk of LSDV transmission by blood feeding insects suggests that *S. calcitrans* and *Aedes aegypti* are the most competent vectors of LSD, whereas *Culicoides nubeculosus*, *Anopheles stephensi* and *Culex quinquefasciatus* are unlikely to be competent at transmitting LSDV [13]. Previous studies have reported the capability of stable flies (*Stomoxys calcitrans*) to transmit sheep pox and goat pox virus [14,15] and LSDV by mosquitoes (*Aedes aegypti*) [16]. In more recent studies, stable flies (*Stomoxys calcitrans*) have been shown to transmit LSDV from experimentally infected animals to naive cattle [15]. It is anecdotally believed that stable flies are most likely involved in LSD epidemiology, since the virus had been recovered from individual flies caught on LSDV-infected animals in the field [10,17]. The aim of this study was to determine the vector competence of three *Stomoxys* spp. in the transmission of LSD virus. 

## 2. Materials and methods

### 2.1. Animal Ethics

Animals were maintained in the insect-free facility with drinking water and hay provided *ad libitium*. The experiment was conducted in accordance with national and international laws based on the European Convention for the Protection of Vertebrate Animals used for Experimental and Other Scientific Purposes guidelines [18]. The protocol was approved by the Committee on the Ethics of Animal Experiments of the Research Institute for Biological Safety Problems of the Science Committee of Ministry of Education and Science of the Republic of Kazakhstan (permit number: 0904/126).

### 2.2. Experimental Animals

Eight Kazakh White-headed cattle, approximately 12 months old, were purchased from a local farm (Korday District) where LSDV has never been recorded and vaccination was not practiced. Prior to the experiment, animals were kept under quarantine conditions in the insect secured containment laboratory of the Research Institute for Biological Safety Issues (RIBSP) for 15 days. During the quarantine period, blood samples were tested for the presence of antibodies against capripox viruses using the serum neutralization test (SNT) [19].

After a veterinarian examination had determined that the animals were healthy, they were divided into 4 groups and placed in isolated rooms of Vivarium No. 202 of the RIBSP.

### 2.3. Experimental Design

Two donor animals (D1 and D2) were artificially infected with a virulent strain of LSDV and served as a source of infection for *Stomoxys calcitrans*, *Stomoxys sitiens* and *Stomoxys indica*.

The LSDV-negative *Stomoxys* spp. colonies used in this study were provided by the Entomology section of the RIBSP. Morphological identification was carried out using Xper2 software (LIS, France) and a protocol based on Zumpt [20].

The titre of the virus inoculum was determined as 10^6^ TCID_50_/mL. The virus suspension was inoculated in a volume of 2 mL at two sites on each side of the body (i.e., neck and flank) and administered subcutaneously. Personnel participating in this experiment wore overalls (DuPont, Tyvek, Wilmington, DE, USA) and respirators. Plastic baits containing flies were disinfested externally using Lysoformin 3000 prior to moving between animal units. Five-day-old unfed batches of *Stomoxys calcitrans*, *Stomoxys sitiens* and *Stomoxys indica* colonies containing approximately 100 flies each were allowed to feed to repletion on the shaved side of the neck and flank of the donor animals 5 days following the development of LSDV lesions or 9 days post infection. The baits were installed manually on the areas with a high concentration of lesions. Engorged flies were then tested for LSDV presence immediately after engorgement and at different time intervals: 6, 24 and 48 h post-feeding. They were maintained in the environmental chamber at 25 °C and 80% humidity, and heparinized bovine blood soaked into cotton pads were offered.

The second batch of newly obtained colonies containing approximately 400 flies each was placed onto an animal with LSDV by using a glass or plastic cylindrical bait containing 40 flies each, closed with a reticular membrane at both ends. Flies on lesions were fed for not more than one minute to prevent the complete engorgement of flies with blood, and then they were moved to a healthy animal to complete feeding. The feeding completion time was determined by the cessation of the flies’ attacking the skin. Recipient animals were observed daily for clinical manifestation of LSDV. Blood samples for PCR and virus isolation were taken at intervals of 3, 5, 7, 9, 11, 13, 15, 17, 19, 21, 23 and 25 days post exposure. Serum samples were collected on day 7, 14, 21 and 28 post exposure.

In addition, all three *Stomoxys* species were examined for the presence of LSDV in the fly proboscis following feeding on donor animals. To do so, new colonies of 100 adult (*S. calcitrans*), 100 (*S. sitiens*) and 150 (*S. indica*) were fed to repletion on lumpy skin disease infected cattle. Following blood–meal engorged flies were anesthetized and kept in wet ice. Flies were then washed three times in PBS and rinsed using distilled water to eliminate surface contamination. Proboscises were removed aseptically using entomological forceps and tested separately according to species. Removed proboscises were homogenized and tested by PCR and VI. 

### 2.4. Virus Strain

The LSDV strain used in this study was isolated from an infected cow during an outbreak of LSD in Atyrau, Kazakhstan in 2016. The virus was passaged five times on primary lamb testis (LT) and three times in vitro using calves. According to Plowright and Ferris [21] pre-pubertal lambs were used to prepare primary LT cell cultures. Cell cultures showed 90% cytopathic effects (CPEs), were freeze–thawed three times, and then centrifuged (2000× *g* for 20 min) and stored at −80 °C until needed. 

### 2.5. Virus Amplification Test

A PCR assay was performed using the protocol published by Tuppurainen, Venter [22].

For DNA extraction, a QIAamp DNA Kit (QIAGEN, USA) was used according to the manufacturer’s instructions. 

For PCR assay, to produce 192 bp of amplified nucleotide reactions, the forward 5’-TCC-GAG-CTC-TTT-CCT-GAT-TTT-TCT-TAC-TAT-3’ and reverse 5’-TAT-GGT-ACC-TAA-ATT-ATA-TAC-GTA-AAT-AAC-3’ primers were used [23]. The conditions for DNA amplification in a Thermal Cycler (Eppendorf Mastercycler) were as follows: 95 °C for 2 min, 95 °C for 45 s, 50 °C for 50 s, 72 °C for 1 min (34 cycles), and 72 °C for 2 min. The PCR products obtained were loaded in 1.5% agarose-gel electrophoresis, and the results visualized using Bio-Imaging Systems MiniBIS Pro (Israel).

### 2.6. Serum Neutralization Test

The serum neutralization test was carried out utilizing 96 well microtitre plates following protocols of the BSL-3 laboratory of the RIBSP based on OIE [24] manuals. Blood samples were collected from experimental animals in vacutainers containing clot activator on days 7, 14, 21 and 28 following infected insect exposure. Serum was separated from clotted blood samples by centrifugation (1500× *g* for 10 min), diluted 1/5 in Eagle’s/HEPES (N2-hydroxyethylpiperazine, N-2-ethanesulphonic acid) and heat inactivated at 56 °C for 30 min. Next, to each well of the 96 well microtitre plates, 50 μL of diluted serum aliquots were added. Ten-fold serial dilution of LSDV from 10^−1^ to 10^−8^ was diluted in Eagle’s/HEPES in bijoux bottles. Fifty microlitres of the virus dilution was introduced to each well in triplicates per dilution, incubated at 37 °C for 1 h before adding LT cells to each well of the plates at a concentration of 10^5^ cells/mL. The monolayers were examined daily for specific CPE and by inverted microscopy on day 9 post inoculation and end-points calculated according to Karber [25]. To determine virus neutralization index, the difference between the titre of virus in the test serum and negative serum from the same animal was taken. A neutralization index showing ≥1.5 was considered positive.

### 2.7. Virus Isolation

Virus isolation was conducted according to standard operational procedures of the BSL-3 laboratory of the RIBSP, based on OIE [24] manuals. Briefly, 1 mL buffy coat or supernatant were administered on to lamb testes cells in 25 cm^2^ cell culture flasks and allowed to incubate at 37 °C for 1 h. Following incubation, culture media was rinsed with PBS and overlaid with Glasgow’s Minimal Essential Medium containing 0.1% penicillin, 0.2% gentamycin and 2% foetal calf serum. The cell monolayer was examined daily for characteristic CPE. In the case no CPE was observed, the cell culture was freeze–thawed three times and the two or three blind passages were carried out. The utilized culture media were kept at −80 °C until required. Cell culture flasks showing CPE were tested with gel-based PCR to confirm that CPE was induced by LSDV.

## 3. Results

As shown in Table 1, PCR and virus isolation tests were positive for all three *Stomoxys* spp. collected immediately and 6 h post-feeding. No virus was isolated from species from 24 and 48 h post-feeding, whereas viral nucleic acid was detected up to 48 h post-feeding by PCR. Further tests failed to detect LSDV in all three species between days 3 and 10 post-feeding. 

### 3.1. Demonstration of LSDV Transmission by Stomoxys calcitrans

Recipient animals R1 and R2 showed characteristic clinical signs of LSD similar to those described by Tuppurainen and Oura [6] and Coetzer [26]. On day 8 post infection, the body temperature for R1 increased to 39.6 °C and reached 40.1 °C on day 17 (Figure 1). In addition to this prescapular and precrural lymph node enlargement and excessive nasal and ocular discharge was observed. Multiple small skin nodules developed on the fly feeding side of R2 and caused pain when palpated, indicated by animals twitching the skin, flicking the tail, kicking or stamping. Elevation in body temperature of R2 increased to 39.9 °C on day 6 post exposure and remained mostly beyond 39.0 °C up to day 28 post exposure (Table 2).

Blood samples collected for PCR tested positive for R1 on days 7, 13, 17 and 21 post exposure. Samples for R2 were positive on days 11, 13 and 15 post exposure. Virus was isolated from blood samples of animal R1 on day 15, whereas samples from R2 exhibited CPE on days 9, 11, 14, 21 and 28 post feeding (Table 3). The index for virus neutralization was ≥log1.5. 

### 3.2. Demonstration of LSDV Transmission by Stomoxys sitiens

On day 6 post infection, the body temperature for R3 was 39.3 °C and elevated up to 39.7 °C on day 13 post exposure (Figure 1). The body temperature for R4 was 39.5 °C on day 9 post exposure and remained chiefly above 39.0 °C up to day 17 post exposure (Table 2). Recipient animals R3 and R4 developed large swellings on the fly feeding sites on day 7 post exposure (Table 2). In addition, the enlargement of prescapular lymph nodes were detected when palpated. On day 14 post exposure R3 developed oedema of the hind limbs extending from metatarsus to tarsal joints (Table 2). R4 manifested nasal and oral discharge as well as rhinitis on day 11 post exposure. Virus DNA was detected from blood samples for R3 by conventional PCR on days 7, 9, 13 and 17 post exposure, while R4 tested positive on days 11 and 15 post exposure. Virus isolation was conducted using a buffy coat, and CPE in the cell culture was seen on the second blind passage for blood samples collected on day 13 post exposure (Table 3). Virus neutralization index demonstrated an index > log1.5 for both recipient animals on days 14 and 21 post exposure.

### 3.3. Demonstration of LSDV Transmission by Stomoxys indica

Animal R5 manifested a mild elevation in body temperature of 39.1 °C on day 10 post exposure and 39.8 on day 12 post exposure (Figure 1). Mild enlargement of prescapular lymph nodes was also detected. Skin lesions, approximately 3 cm in diameter, was observed on feeding sites of flies on day 10 post exposure and disappeared on day 11 post exposure (Table 2). Blood samples tested positive by PCR on day 9 and 11 post exposure. No CPE was seen in the cell culture infected. Seroconversion was detected on days 14 and 21 post exposure (Table 3), showing an index > log1.5.

On the other hand, the recipient animal R6 demonstrated no clinical signs of LSD apart from small-sized swellings on the insect feeding site on day 2 post exposure (Table 2), which gradually disappeared by day 4 post exposure. All samples tested were negative.

### 3.4. Detection of LSDV in the Proboscis of Stable Flies Fed on LSDV-Infected Cattle

The LSDV DNA was detected using gel-agarose PCR in 10 out of 12 samples collected from *S. calcitrans*, in 9 out of 12 samples for *S. sitiens* and in six of the 12 proboscis samples for *S. indica* respectively. The proboscis samples from *S. calcitrans* demonstrated CPE in 14 of 20 (70%) flasks. The samples for *S. sitiens* demonstrated CPE in twelve out of eighteen (66%) flasks while the sample for *S. indica* demonstrated CPE in 7 of 22 (32%) flasks.

## 4. Discussion

There are several studies investigating the role of stable flies as a mechanical vector for the transmission of disease [27,28,29]. It has been assumed that the transmission of LSDV by blood-feeding insects provided a short-term mechanism of transmission [30]. The outcomes obtained in this study indicate a longer duration of possible transmission. In this study, LSDV was shown to survive inside infected stable flies for at least 6 h without a noticeable loss in titre. In other words, the virus could be localized within the insect vector, where stable flies play an intermediary role for virus transmission. In addition to that, harboured LSDV is protected from detrimental ambient conditions, and this implies a more sophisticated means of transportation than just “occasional contact”. 

In the present study, the mechanical transmission of LSDV using *Stomoxys* species (*Stomoxys calcitrans*, *Stomoxys sitiens* and *Stomoxys indica*) from infected to susceptible animals has been demonstrated under laboratory conditions. Five out of six cattle exposed to infected flies refeeding manifested mild to generalized LSD including viremia and fever. Animal R6 did not develop any clinical signs of LSD, apart from small-sized swellings on the insect feeding site on day 2 post exposure. This case could be potentially explained by the differences in the immune status of animals at the moment of exposure. Although the severity of clinical signs exhibited varies in animals, this aligns with records that less than 50% of the cattle infected experimentally or naturally with LSDV will manifest inapparent or generalized disease [31,32]. All three species of flies demonstrated the capability to ingest and harbour virus particles and were able to transmit virus within a 1 h time interval between the feeding processes. Moreover, LSDV was recovered from fly mouthparts within the same period with a virus titre 10^−4^TCID_50_/mL. Furthermore, LSDV can survive in *Stomoxys* species at least 6 h following feeding on an infected animal, whereas viral nucleic acid was detected up to 48 h post exposure. 

In a previous study, stable flies failed to transmit LSDV from infected to naïve animals 24 h after feeding on an infected animal. This implies that virus survival decreases over time in the flies’ gut environment. However, in the transmission experiments with shorter transmission periods, *S. calcitrans* demonstrated a vector competence of transmitting sheep pox and goat pox viruses [33] and LSDV [15]. Stable flies are known to be intrusive feeders, and due to aggressive attacks and painful bites, host animals take defensive actions resulting in interrupted feeding requiring flies to seek for a new host. Such feeding behaviour thus requires taking 3 to 5 interrupted feeding sessions to achieve full repletion [34]. Blood–meal regurgitation by *S. calcitrans* prior taking another blood–meal has been experimentally recorded by Butler and Kloft [35]. This implies that the mouthparts can be contaminated with the virus regurgitated during the second bloodmeal which, in turn, will increase interrupted transmission rates of LSDV by stable flies. 

In a recent study, the mechanical transmission of LSDV by hard ticks has been demonstrated [36,37]. Moreover, they were capable of vertical transmission of LSDV [38,39]. However, the mechanical role of the hard ticks in the mass dissemination of LSDV within a herd is highly restricted since, in most cases, the life cycle of the tick occurs on a single host [40,41]. Taking into account the switch of hosts during stages and generally one meal per stage, their role as even putative vectors diminishes. After repletion, the female ticks drop to the ground, oviposit and die whereas male ticks remain on host animals for further feeding and mating with newly attached females [42]. 

Under field conditions, a large number of flies that feed on the erupted lesions must carry or at least become contaminated with pathogen during the feeding process. Thus, it is highly likely that a single infected animal introduced into a herd will serve as virus source for a large number of resident stable flies. Given the fact that cattle can be easily infected with LSDV using an intradermal method of inoculation leads to the suggestion of insect vector involvement since insects introduce virus into their hosts in a similar manner. This factor will inevitably result in the fulminant spread of the disease within a herd.

It is reported that under experimental conditions stable flies range over large areas, with male flies travelling up to 28.9 km and females 21.9 km in 24 h [43]. Moreover, the wind has a direct impact on insect distribution [44]. In other studies, mathematical modelling used to calculate vector born dispersal of LSDV between herds located in close proximity revealed that most transmission is highly likely to occur over short distances, less than 5 km [45]. Such a significant coverage range and vector capability of stable flies to carry pathogen may lead to LSDV escape from the initial outbreak foci and rapid dissemination over neighbouring farms. Given this fact, insect control programs must be considered during outbreaks of LSD as well as ring vaccination program within the flight range of hematophagous flies.

## 5. Conclusions

The results of this investigation show that *Stomoxys* spp. are ideal vector candidates in the epidemiology of LSD. However, further research is required to establish the retention and persistence of LSDV in *Stomoxys* spp. following per os or intrathoracic inoculation.

## Figures and Tables

**Figure 1 animals-10-00477-f001:**
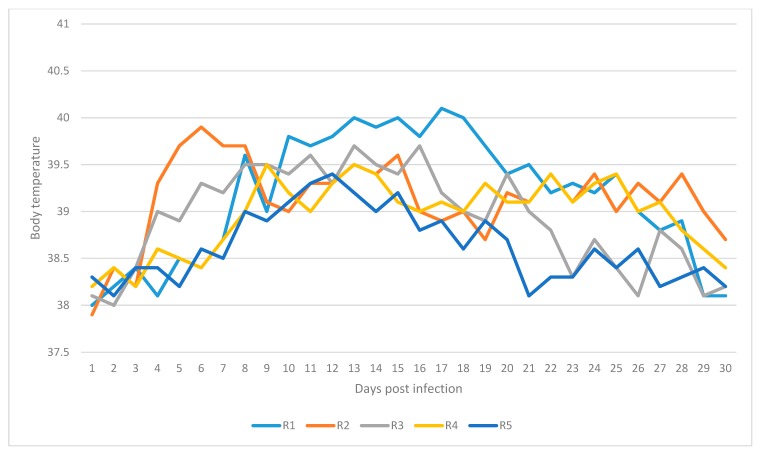
Daily temperatures for recipient animals: R1, R2, R3, R4 and R5.

**Table 1 animals-10-00477-t001:** PCR and virus isolation results of *Stomoxys* spp. at different time intervals following feeding on lumpy skin disease virus LSDV infected donor animals.

Time/Day Post Feed	*Stomoxys calcitrans*			*Stomoxys sitiens*			*Stomoxys indica*		
	PCR (No Positive/No Tested)	Virus Isolation (No Positive/No Tested)	Virus Isolation (TCID_50_/mL), Mean	PCR (No Positive/No Tested)	Virus Isolation (No Positive/No Tested)	Virus Isolation (TCID_50_/mL), Mean	PCR (No Positive/No Tested)	Virus Isolation (No Positive/No Tested)	Virus Isolation (TCID_50_/mL), Mean
0 h	12/15	9/15	3.5	13/15	12/15	3.1	11/15	10/15	3.3
6 h	8/15	4/15	1.8	11/15	5/15	2.0	8/15	4/15	1.9
1	5/15	0/15	-	4/15	0/15	-	3/15	0/15	-
2	2/15	0/15	-	3/15	0/15	-	2/15	0/15	-
4	0/15	0/15	-	0/15	0/15	-	0/15	0/15	-
7	0/15	0/15	-	0/15	0/15	-	0/15	0/15	-
14	0/15	0/15	-	0/15	0/15	-	0/15	0/15	-

**Table 2 animals-10-00477-t002:** Clinical manifestations of LSD for recipient animals infected through transmission of LSDV by *Stomoxys* spp.

Animal ID	Route of Transmission	Numbers and Species of Flies Fed	Body Temperature	Lesions	Enlargement of Lymph Nodes	Additional Clinical Signs
R1	Mechanical	210 *Stomoxys calcitrans*	Up to 40.1 °C (on day 17 post exposure)	Small skin nodules developed on fly feeding sites (Day 10 post exposure)	Mild enlargement: prescapular and precrural lymph nodes (Day 11 post exposure)	Nasal and ocular discharges (Day 11 post exposure)
R2	Mechanical	210 *Stomoxys calcitrans*	Up to 39.9 °C (on day 6 post exposure)	Small swellings at feeding sites (Day 14 post exposure)	Mild enlargement: prescapular and precrural lymph nodes (Day 9 post exposure)	Nasal and ocular discharges (on day 6 post exposure)
R3	Mechanical	200 *Stomoxys sitiens*	Up to 39.7 ℃ (on day 13 post exposure)	Large swellings at feeding sites (Day 7 post exposure)	Mild enlargement: prescapular lymph node (Day 13 post exposure)	Oedema of the hind limbs (Day 13 post exposure)
R4	Mechanical	200 *Stomoxys sitiens*	Up to 39.5 ℃ (on day 9 post exposure)	Skin lesions at feeding sites (Day 14 post exposure)	Mild enlargement: prescapular and precrural lymph nodes (Day 14 post exposure)	Nasal and oral discharges, rhinitis (Day 11 post exposure)
R5	Mechanical	230 *Stomoxys indica*	Up to 39.8 ℃ (on day 12 post exposure)	Small skin lesions at feeding sites (Day 10 post exposure)	Mild enlargement: prescapular lymph node (Day 11 post exposure)	-
R6	Mechanical	230 *Stomoxys indica*	-	Small-sized swellings on the feeding site (Day 2 post exposure)	-	-

**Table 3 animals-10-00477-t003:** Test results obtained from blood samples of animals R1, R2, R3, R4, R5 and R6.

Animal ID	Transmission	Gel-Based PCR	SNT	Virus Isolation/Titre	Results
R1	Mechanical	Positive (days 7, 13, 17, 21 pe)	Positive (days 14, 21, 28 pe)	Positive (day 15 pe)/10^3^ TCID_50_	Infected
R2	Mechanical	Positive (days 11, 13, 15 pe)	Positive (days 14, 21, 28 pe)	Positive (days 9, 11 pe)/10^2.5^ TCID_50_	Infected
R3	Mechanical	Positive (days 7, 9, 13, 17 pe)	Positive (days 14, 21 pe)	Positive, 2nd passage (day 13 pe)/ 10^3.5^ TCID_50_	Infected
R4	Mechanical	Positive (days 11, 15 pe)	Positive (days 14, 21 pe)	Positive, 2nd passage (day 13 pe)10^3^ TCID_50_	Infected
R5	Mechanical	Positive (day 9, 11 pe)	Positive (days 14, 21 pe)	-	Infected
R6	Mechanical	-	-	-	Not Infected

(pe)—post exposure.

## Data Availability

The data obtained during this study are openly available in KNB Data Repository at http://doi.org/10.5063/F1N29V8C.

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
