# Peer review of "Mechanical Transmission of Lumpy Skin Disease Virus by Stomoxys spp. (Stomoxys calsitrans, Stomoxys sitiens, Stomoxys indica), Diptera: Muscidae"

_animals, 2020, doi:10.3390/ani10030477_

Round 1
Reviewer 1 Report
Issimov et al. investigate the competence of Stomoxys spp. as potential vectors of Lumpy skin disease. It does not add a lot to the data study recently published by Sohier et al. Maybe the only new finding here is that various species of Stomoxys (and not just S. calcitrans), can transmit the virus. On the other hand, it seems that in this study, Stomoxys do succeed in infecting the cattle, but in all the cases it is not clear whether a real disease is developed. As I understand, none of the cows developed generalized disease. Documenting the rectal temperature of the infected cattle every day may show if the infected cows really developed a long-lasting fever. Generally, I think that this study adds important data on the subject. However, in my opinion, prior to its publication it should be improved by writing a more detailed description of the materials and methods and the results as well as performing a better review of the literature.
Lines 41-43: This is also based on experiments showing no direct transmission of LSD (see Carn VM, Kitching RP. An investigation of possible routes of transmission of lumpy skin disease virus (Neethling). Epidemiol Infect. 1995 Feb;114(1):219-26.) and on mathematical modeling of transmission during an outbreak (see Magori-Cohen R, Louzoun Y, Herziger Y, Oron E, Arazi A, Tuppurainen E, Shpigel NY, Klement E. Mathematical modeling and evaluation of the different routes of transmission of lumpy skin disease virus. Vet Res. 2012 Jan 11;43:1.).
Lines 50-52: The assumption regarding the transmission of LSD by S. calcitrans is also based on its temporal abundance, which was associated with the occurrence of LSD outbreaks (see Kahana-Sutin E, Klement E, Lensky I, Gottlieb Y. High relative abundance of the stable fly Stomoxys calcitrans is associated with lumpy skin disease outbreaks in Israeli dairy farms. Med Vet Entomol. 2017 Jun;31(2):150-160. ).
Line 53: Change 'capability' to 'competence'.
Lines 73-84: It is not clear when exactly the flies were fed and how was this performed. Was it after the development of clinical signs by the donors, how long after experimental infection? On what site of the infected cattle? For how long were the flies fed? What is the source of the flies (colonies or captured in the field?).
Table 2: R1 lines under 'lesions': change 'side' to 'site'.
Lines 158-165: Adding a figure showing the body temperature in each day of the study for each animal, will be very helpful. The same goes for all other experiments mentioned later.
Line 179: change 'buffer' to 'buffy'.
Table 3: Please add 'blood' in the title.
Please elaborate, what was the virus concentration in the blood?
Author Response
Thank you for providing a review for manuscript. We have addressed issues in the comments and suggestions. Please see the attachment.
Point 1: Lines 41-43 This is also based on experiments showing no direct transmission of LSD (see Carn VM, Kitching RP. An investigation of possible routes of transmission of lumpy skin disease virus (Neethling). Epidemiol Infect. 1995 Feb;114(1):219-26.) and on mathematical modeling of transmission during an outbreak (see Magori-Cohen R, Louzoun Y, Herziger Y, Oron E, Arazi A, Tuppurainen E, Shpigel NY, Klement E. Mathematical modeling and evaluation of the different routes of transmission of lumpy skin disease virus. Vet Res. 2012 Jan 11;43:1.).
Response 1: References have been added.
Point 2: Lines 50-52 The assumption regarding the transmission of LSD by S. calcitrans is also based on its temporal abundance, which was associated with the occurrence of LSD outbreaks (see Kahana-Sutin E, Klement E, Lensky I, Gottlieb Y. High relative abundance of the stable fly Stomoxys calcitrans is associated with lumpy skin disease outbreaks in Israeli dairy farms. Med Vet Entomol. 2017 Jun;31(2):150-160. ).
Response 2: References have been added.
Point 3: Line 53 Change 'capability' to 'competence'.
Response 3: The word “capability” has been changed to “competence”.
Point 4: Lines 73-84 It is not clear when exactly the flies were fed and how was this performed. Was it after the development of clinical signs by the donors, how long after experimental infection? On what site of the infected cattle? For how long were the flies fed? What is the source of the flies (colonies or captured in the field?).
Response 4: Flies on lesions were fed for not more than one minute to prevent the complete engorgement of flies with blood, and then they were moved to a healthy animal to complete feeding. Stable flies used in this study were stock colony reared at Entomology section. Neck and flank sites were used for LSDV inoculation in donor animals, whereas in recipient animals they were used as flies feeding sites. The baits were installed manually on the areas with high concentration of lesions. Fleis were allowed to feed on the shaved side of the neck and flank of the donor animals 5 days following the development of LSDV lesions or 9 days post infection.
Point 5: Table 2: R1 lines under 'lesions' change 'side' to 'site'.
Response 5: Revised as requested.
Point 6: Lines 158-165 Adding a figure showing the body temperature in each day of the study for each animal, will be very helpful. The same goes for all other experiments mentioned later.
Response 6: Lines 158-165 Figures of daily rectal temperature have been added.
Point 7: Line 179 change 'buffer' to 'buffy'.
Response 7: Revised as requested.
Point 8: Table 3 Please add 'blood' in the title. Please elaborate, what was the virus concentration in the blood?
Response 8: Word “blood” added in the title and virus concentration has been elaborated.

Reviewer 2 Report
The study described by Issimov et al provides an evidence that three Stomoxys species can successfully transmit LSDV between infected and naive cattle.
a few minor changes:
line 27: change to "LSDV belongs to the Capripoxvirus genus of the Poxviridae family.
Abstract and text: use either hour or h to be consistent.
Lines 73, 79-80, 90-92: change font type and size to be consistent.
Line 91: change "VI" to virus isolation
line 105: "-80" degree sign is missing
Line 188: "no CPE was seen in the blood samples collected" - incorrect sentence. CPE can be seen in the cell culture
line 223: Is a virus titre 10^-4 TCID50 correct?
line 251: "with male flies travelling up to 28.9 km and females 21.9 km" - can you specify the time period. Can they fly this distant within 1 hour, 2 hours? etc.
line 48: sheep pox and goat pox (12,13) - spaces are missing
Author Response
Thank you for providing a review for manuscript. We have addressed issues in the comments and suggestions. Please see the attachment.

Reviewer 3 Report
The manuscript is interesting and the results are globally well presented.
Main issue is the lack of any information about the flies: i) where they came from (field or insect facility)? How the species has been determined? Were they LSDV negatives at the beginning of the experiment? and how it can be demonstrated? All these information should be provided in order to evaluate if the manuscript worth to be published.
Materials and Methods section should be improved, notably:
- line 72, paragraph 2.3. “Experimental Design” in the present form, is not clear and is sometime misleading. The experimental design should be re-written in order to give a comprehensive and schematic description of the experiment.
- line 78: manuscript mentions respirators (class N95). The reason for using such device should be explained, otherwise is better to remove this detail.
- lines 93-99, it is not clear which flies were tested for the presence of LSDV in proboscis, whether they belong to the same batches or to others. Please re-write.
- line 103, there is probably a mistake in the sentence “three times in vitro using calves”.
- lines 109-116, since this paragraph report a commercial kit protocol for DNA extraction, it could be removed.
- line 179, “buffer coat” should be probably replaced by buffy coat
Table 1, the number of positive samples for virus isolation should be reported for the three fly species
Table 3 animal R2 was positive in VI, before than PCR. This result should be explained.
Author Response
Point 1: Main issue is the lack of any information about the flies: i) where they came from (field or insect facility)? How the species has been determined? Were they LSDV negatives at the beginning of the experiment? and how it can be demonstrated? All these information should be provided in order to evaluate if the manuscript worth to be published.
Response 1: LSDV negative Stomoxys spp colonies used in this study were provided by Entomology section of the RIBSP. Morphological identification was carried out using Xper2 software (LIS, France) and protocol based on Zumpt.
Point 2: Line 72, paragraph 2.3. “Experimental Design” in the present form, is not clear and is sometime misleading. The experimental design should be re-written in order to give a comprehensive and schematic description of the experiment.
Response 2: Paragraph 2.3. “Experimental Design” has been re-written.
Point 3: Line 78: manuscript mentions respirators (class N95). The reason for using such device should be explained, otherwise is better to remove this detail.
Response 3: (class N95)” has been removed.
Point 4: Lines 93-99, it is not clear which flies were tested for the presence of LSDV in proboscis, whether they belong to the same batches or to others. Please re-write.
Response 4: Additional information has been added. “In addition, all three Stomoxys species were examined for the presence of LSDV in the fly proboscis following feeding on donor animals. To do so, a new colonies of 100 adult (S. Calcitrans), 100 (S. sitiens) and 150 (S. indica) were fed to repletion on lumpy skin disease infected cattle”.
Point 5: Line 103, there is probably a mistake in the sentence “three times in vitro using calves”.
Response 5: This implies that LSDV was passaged three times using alive young bovine animal to make a virus more virulent.
Point 6: Lines 109-116, since this paragraph report a commercial kit protocol for DNA extraction, it could be removed.
Response 6: The paragraph containing commercial kit protocol for DNA extraction has been removed
Point 7: Line 179, “buffer coat” should be probably replaced by buffy coat
Response 7: Revised as requested.
Point 8: Table 1, the number of positive samples for virus isolation should be reported for the three fly species
Point 9: Table 3 animal R2 was positive in VI, before than PCR. This result should be explained.
Reviewer 4 Report
The paper is well written. I have no objections to the publication and the described results. I only suggest write a small summary (conclusion) at the end of discussion.
Author Response

(The authors gave the same response as above.)

Round 2
Reviewer 3 Report
The modified version of the manuscript is suitable for publication. I suggest two additional minor changes
Point 3: Line 78, since LSDV is neither zoonotic nor spreading through the air, is not clear why respirators were used during experimental activity.
Point 5: Line 103, the meaning is clear, but in vitro should be replaced by in vivo.
Author Response

(The authors gave the same response as above.)
